# Metagenomic Insights and Genomic Analysis of Phosphogypsum and Its Associated Plant Endophytic Microbiomes Reveals Valuable Actors for Waste Bioremediation

**DOI:** 10.3390/microorganisms7100382

**Published:** 2019-09-23

**Authors:** Fedia Ben Mefteh, Ali Chenari Bouket, Amal Daoud, Lenka Luptakova, Faizah N. Alenezi, Neji Gharsallah, Lassaad Belbahri

**Affiliations:** 1NextBiotech, 98 Rue Ali Belhouane, Agareb 3030, Tunisia; fedia.benmefteh@gmail.com (F.B.M.); amal.daoud42@gmail.com (A.D.); Lenka.Luptakova@uvlf.sk (L.L.); Dr-Faizah@outlook.com (F.N.A.); 2Faculty of Sciences, University of Sfax, Sfax 3029, Tunisia; neji.gharsallah@fss.rnu.tn; 3Plant Protection Research Department, East Azarbaijan Agricultural and Natural Resources Research and Education Center, AREEO, Tabriz 5355179854, Iran; a.chenari@areeo.ac.ir; 4Department of Biology and Genetics, Institute of Biology, Zoology and Radiobiology, University of Veterinary Medicine and Pharmacy in Košice, 04181 Kosice, Slovakia; 5Laboratory of Soil Biodiversity, University of Neuchâtel, CH-2000 Neuchatel, Switzerland

**Keywords:** phosphogypsum, endophytic bacteria, metagenomic analysis, metal resistance, industrial effluents

## Abstract

The phosphogypsum (PG) endogenous bacterial community and endophytic bacterial communities of four plants growing in phosphogypsum-contaminated sites, *Suaeda fruticosa* (*SF*), *Suaeda mollis* (*SM*), *Mesembryanthmum nodiflorum* (*MN*) and *Arthrocnemum indicum* (*AI*) were investigated by amplicon sequencing. Results highlight a more diverse community of phosphogypsum than plants associated endophytic communities. Additionally, the bacterial culturable communities of phosphogypsum and associated plant endophytes were isolated and their plant-growth promotion capabilities, bioremediation potential and stress tolerance studied. Most of plant endophytes were endowed with plant growth-promoting (PGP) activities and phosphogypsum communities and associated plants endophytes proved highly resistant to salt, metal and antibiotic stress. They also proved very active in bioremediation of phosphogypsum and other organic and inorganic environmental pollutants. Genome sequencing of five members of the phosphogypsum endogenous community showed that they belong to the recently described species *Bacillus albus* (*BA*). Genome mining of *BA* allowed the description of pollutant degradation and stress tolerance mechanisms. Prevalence of this tool box in the core, accessory and unique genome allowed to conclude that accessory and unique genomes are critical for the dynamics of strain acquisition of bioremediation abilities. Additionally, secondary metabolites (SM) active in bioremediation such as petrobactin have been characterized. Taken together, our results reveal hidden untapped valuable bacterial actors for waste remediation.

## 1. Introduction

When using rock phosphate as raw material for the production of phosphoric acid, phosphogypsum (PG) is the main waste generated. Phosphogypsum, currently considered a NORM (naturally occurring radioactive material), contains numerous hazardous materials such as natural radionuclides and heavy metals [1]. Management of phosphogypsum is challenging in numerous countries. In Tunisia, when discharged on the marine environment in the Gulf of Gabes, phosphogypsum severely impacts the ecosystem where environmental conditions can be considered as critical [2]. Its use as a fertilizer, while environmentally and economically sound, is hindered by its high toxicity to crops and earthworms [3,4,5,6]. Therefore, there is an urgent need to remediate the millions of tons of PG that weaken ecosystem functions and represent serious threats to the environment and human health.

While traditional mitigation technologies failed to treat the high volumes of waste, a more sustainable and eco-friendly mean consists of the use of the PG microbiome able to degrade PG [7,8].

Plants also form associations with diverse beneficial microorganisms that can provide them with selective benefits [9,10,11,12,13,14,15]). Among them, bacterial endophytes can support plant growth, prevent plant diseases and alleviate abiotic stresses [16,17,18,19,20]). Considerable interest is therefore drawn actually to describe endophytic communities from plants that are subjected to diverse types of stress [21,22,23,24]. Under contaminated environments where plants thrive, endophytes can alleviate pollutant-induced stress by producing indole-3-acetic acid (IAA), a plant hormone that promotes their growth and development [10,25]. Endophytes can also solubilize phosphate and nitrogen, which are considered among the elements most limiting to plant growth and development [13,15]. They are also able to release 1-aminocyclopropane-1-carboxylate (ACC) deaminase, responsible for a relevant decrease of ethylene production [10]. It is believed that plants thriving with endophytes endowed with ACC deaminase activity have longer roots and shoots. They are also more resistant to growth inhibition by a variety of ethylene-inducing stresses [21]. HCN production by endophytes is beneficial for plant hosts as suggested by Rijavec and Lapanje [26] through the sequestration of metals and the consequential indirect increase of nutrient availability. Siderophore production by endophytes enables PGP indirectly and are, therefore, considered beneficial for plants thriving under environmental pollution [27].

Taken together, all these elements lead to the proposal that endophytes endowed with PGP potential and contaminant-degrading abilities would be optimal means for the full development of bio- and phyto-remediation to efficiently clean up wastes and contaminants [21].

In this study, to unravel the potential of use of the phosphogypsum microbiome and the potential of endophytic plants thriving in this waste for its remediation, we investigated through cultivation independent (using amplicon sequencing technology) phosphogypsum bacterial microbiome as well as bacterial endophyte microbiomes of four plants *SF*, *SM*, *MN* and *AI* thriving in this waste. Simultaneously, the culturable community was also recovered and their PGP potential, stress tolerance and bioremediation potential were studied. Finally, genome mining of five phosphogypsum endogenous bacteria allowed us to shed the light on their stress tolerance and bioremediation potential at the genomic level.

## 2. Materials and Methods

### 2.1. Phosphogypsum, and Effluents Collection

Phosphogypsum was obtained from the site of company phosphate Gafsa (CPG) located 6 km south of Gafsa (commune of Mdhilla) in Tunisia (34°10′ North 8°45′ East). Phosphogypsum samples were collected on the bottom of the stock piles covering 36 ha with approximately 47,720,000 m^3^ of PG (Figure 1A,B). The samples were transported in sterile bags to the laboratory. The textile wastewater (TWW) samples were obtained from YOUSSTEX international (https://www.yousstex.com/) located in Touza, Tunisia. The TWW effluent was the result of the indigo jean dyeing process. The oil wastewater (OWW) was obtained from AMAL Services and Risks Management Company. AMAL company, located in Sfax Tunisia, is specialized in hazardous waste processing, environment protection and safety for oil related industrial activities.

### 2.2. SF, SM, MN and AI Plant Materials Collection

Plants (*SF*, *SM*, *MN* and *AI*) thriving in the immediate vicinity of the phosphogypsum piles have been collected, placed in sterile plastic bags and transported to the laboratory (Figure 1C–F). Plant materials were separated to roots and shoots immediately after arrival to the laboratory and used for DNA extraction or culturable endophyte community isolation.

### 2.3. DNA Extraction from Phosphogypsum and Plant Materials and Metagenomic Analysis

All procedures followed closely recommendations of Comeau et al. [28] and are detailed in Appendix A and methods section. PICRUSt software package designed to predict metagenome functional content from marker gene (e.g., 16S-rDNA) surveys and full genomes [29], have been used to unravel the functional aspects of bacterial phosphogypsum and plant endophytic microbiomes.

### 2.4. Isolation of Cultivable Bacterial Microbiome of Phosphogypsum and AI, SF, SM and MN Plants

All procedures have been conducted according to Slama et al. [15] and are detailed in the Appendix A and methods section. The microbiological characterization of PG bacterial microbiome was performed by the serial dilution procedure. Briefly, PG dilutions in sterile water (10^−1^, 10^−2^, 10^−3^, 10^−4^, 10^−5^ and 10^−6^) were spread aseptically on the surface of plate count agar (PCA) plates. Plates were incubated at 37 °C for 24 h and colonies with different morphological appearance were selected from the plates and purified by further sub-culturing. The PG and endophytic plant microbiomes, respectively, have been screened in vitro for bioremediation potential and stress tolerance. Additionally, plants endophytic microbiomes have been examined in vitro for PGP activities.

### 2.5. Measurement of PGP Activities of Plant Endophytic Communities

PGP activities studied include phosphate solubilization, Nitrogen fixation, HCN production, ACC deaminase activity and Siderophore Production. All tests have been performed as described in Slama et al. [15] and are detailed in Appendix A and methods section.

### 2.6. Screening of Phosphogypsum and Plant Endophytic Bacterial Microbiomes for Antibiotic, Metal and Salinity (NaCl) Resistance, Pesticides and Effluents Degradation

Screening of the different resistances have been conducted according to Slama et al. [15] and is detailed in the Appendix A and methods section.

### 2.7. Bacterial Genome Sequencing, Assembly and Annotation

Genome sequencing have been performed according to Slama et al. [15] and is detailed in the Appendix A and methods section. Briefly, library was generated using the Nextera DNA Flex Library Prep Kit and sequenced using Illumina MiSeq procedures.

### 2.8. Selection and Phylogenomic Analysis of Phosphogypsum Bacterial Isolates

Phylogenomic analysis have been performed according to Slama et al. [15] and is detailed in the Appendix A and methods section.

### 2.9. Homology-based Mining of Genes Contributing to Detoxification of Organic Pollutants, Competition, Fitness and Comparative Genomics Analysis of BA Isolates

All analyses have been performed according to Slama et al. [15] and are detailed in the Appendix A and methods section.

### 2.10. Secondary Metabolite Clusters Identification Using antiSMASH, NapDos, NP.Search, and Bagel3

Secondary metabolite gene clusters analysis of the annotated draft genome sequences of the *BA* isolates collection was performed using antiSMASH 3.0 [30], NapDos [31], NP.search [32], and the bacteriocin-specific software BAGEL3 [33].

### 2.11. Statistical Analysis

The statistical analysis program used in the time course of the study was IBM SPSS Statistics v. 22 (Geneva, Switzerland). Analysis of variance (ANOVA) was used to perform statistical analysis of the samples. Once a significant effect was detected, the groups were compared using a post hoc Tukey’s HSD test. The level of significance applied for all statistical tests was 5% (*p* < 0.05).

## 3. Results

### 3.1. Physicochemical Characteristics of Phosphogypsum

Phosphogypsum, characterized by a dry weight of 56.9 mg, analysis using ICP-OES revealed high amounts of sodium (5435.389 mg/kg), calcium (3664.460 mg/kg), iron (1103.425 mg/kg), aluminum (790.107 mg/kg), magnesium (490.411 mg/kg) and potassium (313.318 mg/kg). It shows also the presence of strontium (163.470 mg/kg), zinc (135.876 mg/kg), chrome (37.286 mg/kg), cadmium (17.177 mg/kg), barium (14.803 mg/kg), selenium (9.217 mg/kg), copper (6.144 mg/kg), nickel (6.005 mg/kg) and manganese (5.446 mg/kg). Rb, Pb, Li and As with concentrations lower than 3 mg/kg (Appendix A).

### 3.2. Metagenomic Analysis of Phosphogypsum and Plant Endophytic Microbiomes

Figure 2A–C and Appendix A illustrate metagenomic analysis of phosphogypsum and plant endophytic microbiomes using 16S-rDNA amplicon next-generation sequencing. Results unambiguously document dominance of phosphogypsum communities by firmicutes (39.3%), proteobacteria (24.2%) and actinobacteria (20.4%). For phosphogypsum microbiome bacilli (38.3%) dominated the firmicutes with main genera *Bacillus* (20.7%) and *Enterococcus* (17.6%). Alpha proteobacteria (18.8%) dominated the proteobacteria and actinobacteria were dominated by the genus *Kocuria* (15.2%). The plants SF, SM, MN and AI endophytic microbiomes were dominated by protebacteria. SF and SM microbiomes were dominated by Alpha proteobacteria (85.9% and 95.2%, respectively), while MN and AI microbiomes were dominated by Gamma proteobacteria (96.6% and 95.5% respectively) with main genera being Enterobacteriacae (50.9%) undescribed genera and *Halomonas* (44.8%).

High specificity of taxa to origin or host was observed (136), while only two taxa were common to phosphogypsum and plant microbiomes (Figure 3A). Network analysis revealed two main groups of co-occurrences: a first group represented by *Clostridium gasigenes*, *Prevotella copri*, *Cesiribacter adanamensis* and *Shewanella algae* and a second group represented by *Paenibacillus aminolyticus*, *Carnobacterium viridans*, *Bacillus formainis*, *Pseudomonas fragi* and *Chromohalobacter salixigens* (Figure 3B).

Figure 4 and Appendix A summarize common bacterial taxonomic groups between phosphogypsum and endophytic plant microbiomes. While the phosphogypsum microbiome was very diverse, plant endophytic microbiomes were less diverse with few taxonomic groups dominating (Figure 4A–C).

### 3.3. Metagenomic Functional Content of Phosphogypsum and Plant Endophytic Microbiomes

There was no clear difference between phosphogypsum microbiome and SF, SM, MN and AI microbiomes (Figure 5A). For all metagenomes, the main functions were related to the category’s metabolism (49.4%), genetic information processing (16.3%), unclassified (14.8%), environmental information processing (14.7%), cellular processes (3%) and human diseases (1.1%), where 0.7% of them were dedicated to infectious diseases and organismal systems (0.8%). Within the metabolism categories, energy, carbohydrate, amino acid and vitamins and cofactors (5.5%) metabolism were particularly represented. Meanwhile, xenobiotic biodegradation and metabolism (2.3%), metabolism of terpenoids and polyketides (2%) and biosynthesis of other SM (1%) are also represented. Xenobiotics biodegradation and metabolism represented a rich arsenal targeting a diverse class of molecules involving atrazine, benzoate, bisphenol, dioxin, drugs, naphthalene, toluene and xylene among others. Biosynthesis of other SM involved the biosynthesis of antibiotics such as streptomycin, novobiocin and neomycin. For genetic information processing, “replication and repair” was the most represented category (7.1%). For environmental information processing, membrane transport (12.1%) was the most represented and within this category, the main classes are transporters and ABC transporters (6% and 3.7%, respectively). Within cellular processes, cell motility was the most represented (2.2%, Figure 5B).

### 3.4. Phylogenetic Analysis of Cultivable Microbiomes of Phosphogypsum and Plant Endophytes

Culturable phosphogypsum microbiome was dominated by *Bacillus* spp. (Figure 6, Appendix A). Phosphogypsum *Bacillus* spp. belonged mainly to the *Bacillus cereus* group (BC group, Appendix A). They share over 97% similarity with the known species of this group. Culturable plant endophytic communities were more diverse than the phosphogypsum culturable microbiome and lumped representants of 11 different genera. While the BC group was dominating culturable plant endophyte populations (representing more than 60% of each population), each plant has its own specific genera, SF (*Enterococcus*, *Pantoea*, *Serratia*), SM (*Staphylococcus*, *Enterococcus*, *Brevibacillus*, *Pantoea*, *Pseudomonas*, Uncultured bacterium) MN (*Enterococcus*, *Pantoea*) and AI (*Enterococcus*, *Staphylococcus*, *Sphingobium*, *Pantoea*, *Stenotrophomonas* and *Paenibacillus*) (Figure 6A,B, and Appendix A).

### 3.5. Characterization of Phosphogypsum Bacterial and Plant Endophytic Microbiomes for PGP Activities, Bioremediation Potential and Stress Tolerance

More than 80% of the bacterial PG microbiome and plant endophytic microbiomes were able to cope with salt stress induced by the addition of 1M NaCl in the growth medium. However, at 2M NaCl, endophytic communities of SF and MN failed to grow on this medium. Only 38% of the endophytic community of AI were able to grow on NaCl (2M), while 85 and 100% of the endophytic community of bacterial PG microbiome and endophytes of SM were able to grow on NaCl (2M) (Figure 7). More than 90% of the bacterial phosphogypsum microbiome and plant endophytic microbiomes were able to tolerate 100 ppm of the metals Mg, Cd, Al, Pb, Fe, Zn, Cu, Ni and V. For Hg, a minimum of 55% of the bacterial PG and plant endophytic microbiomes were able to tolerate it (Figure 7). Bacterial PG and plant endophytic microbiomes showed varying levels of resistance to Streptomycin (more than 79%), Tetracycline (more than 42%), Rifamycin (more than 39%), Ampicillin (more than 22%) and Penicillin G (more than 10%) (Figure 7).

Regarding biodegradation potential, insecticides imidacloprid, permethrin and dimethoate as well as the herbicide glyphosate were efficiently metabolized by more than 70% of the isolates, when used as a sole nitrogen source, depending on the bacterial PG or plant microbiomes. TWW was efficiently metabolized, when used as a sole nitrogen source at 40% or 60%, by bacterial phosphogypsum and endophytic plant communities of SF, SM, MN and AI (90–30, 70–50, 100–98 and 100–100, respectively, Figure 7). OWW was also efficiently metabolized, when used as a sole nitrogen source at 40% or 60%, by phosphogypsum and plant endophytic communities of SF, SM, MN and AI (90–10; 50–50; 100–98 and 100–100, respectively; Figure 7). Phosphogypsum was also highly used as a sole nitrogen source at 40% or 60%, by phosphogypsum and plant endophytic microbiomes of SF, SM, MN and AI (90–10; 81–93; 100–98; 98–98 and 78–79, respectively; Figure 7).

Among the collection ACC deaminase activity, production of siderophores and phosphate solubilization were the most represented PGP activities. With the exception of SM phosphate solubilization, present in 10% of the isolates, these PGP related activities were present in more than 70% of the plant species endophytic isolates (Figure 7). HCN production and nitrogen fixation were present to a lesser level among the plant endophytic isolates. HCN production activity was present in 42%, 55%, 76% and 32% of the endophytes of SF, SM, MN and AI, respectively. While nitrogen fixation was present in 35%, 63%, 57% and 60% of SF, SM, MN and AI endophytes, respectively (Figure 7).

### 3.6. Identity and Phylogenomic Positions of Bacterial Phosphogypsum Isolates PG 1, PG 9, PG 17, PG 18 and PG 26

Genomic analysis of phosphogypsum bacteria PG 1, PG 9, PG 17, PG 18 and PG 26 allowed the identification of these different isolates as BA. Genome-to-Genome Distance (GGD) analysis and Average Nucleotide Identity (ANI) analysis were both in agreement that PG 1, PG 9, PG 17, PG 18 and PG 26 represent isolates of the species BA. After checking in the GenBank genome repository, two additional genomes of BA (CP034548.1 for strain PFYN01 and MAOE00000000.1 for strain N35-10-2) were retrieved and used for subsequent phylogenomic analysis to ascertain their exact phylogenomic position (Figure 8). Genome size range within BA isolates was from 4.9 to 5.9 Mbps, while GC content ranged from 35.0% to 35.3% (Appendix A). GGD analysis, where 70% similarity between two genomes was established as a suitable cut-off and the gold standard threshold for species boundaries, revealed that all isolates PG 1, PG 9, PG 17, PG 18, PG 26, PFYN01 and N35-10–2 represented the same species sensu Meier-Kolthoff et al. [34] (Figure 8B). ANI analysis also confirmed their single species status sensu Richter and Rossello-Mora [35], where a 95–96% cut-off was set up to delimit species boundaries and confirmed, therefore, the observations drawn using GGD analysis (Figure 8C). Th final aligned genomic datasets contained 26 ingroup taxa with a total of 1642 characters, containing 72 unique site patterns. *Bacillus subtilis* subsp. *subtilis* strain ATCC 6051-HGW (GenBank accession CP003329.1) served as the outgroup. The heating parameter was set to 0.15. The results of MrModeltest v.2.3 recommended JC model with a gamma distributed rate variation and dirichlet base frequencies. During the generation of the tree, a total of 492 trees were saved, and consensus trees and posterior probabilities were calculated from the remaining 370 (75%) trees. Whole genome phylogeny was also in agreement with earlier results and documented the lumping of the seven isolates in BA (Figure 8A).

### 3.7. Characterization of the Core and Pan Genome of BA

Comparative genomics analysis of PG 1, PG 9, PG 17, PG 18, PG 26, PFYN01 and N35-10-2 document that they represent different strains of same species (Figure 9). Differences are visible among strains (Figure 9A, Figure 10C). These differences have been further confirmed by synteny analysis and by analysis of conserved and unique gene families of the strains (Figure 9B,C). Pan and core genomes of given species are nowadays required to shed the light and explore the whole genomic and metabolic potentialities of the species isolates full range [10,15]. Therefore, BA available genomes were mined for pan and core genomes using bioinformatic tools BPGA [36], Spine and Agent [37]. Figure 10A,B document that the BA pan genome increased with the number of new genomes. Heaps’ law model fitted to the number of new gene clusters observed when genomes are ordered in a random way, as implemented in Tettelin et al. [38], is in agreement with an open BA pan genome that increases with α values of 0.05.

### 3.8. Functional Characterization of the BA Core, Accessory and Unique Genomes

The COG distributions of BA pan, accessory and unique genomes were generated (Figure 11A). There were notable differences between the genomes. While core genome was enriched in basic metabolic functions as translation, ribosomal structure and biogenesis (J) and coenzyme and nucleotide transport and metabolism (H), functional genes related to secondary metabolite biosynthesis, transport, and catabolism (Q) and defense mechanisms (V) have been enriched in the accessory genome. A unique genome was however enriched in cell wall, membrane and envelope biogenesis (M) and replication recombination and repair (L) along with genes of unknown or general function prediction only (R and S). BA KEGG pathway analysis of metabolic features allowed to unravel the distribution of these features in the three genomes. While energy, amino acids, cofactors and vitamins metabolism were particularly enriched in the core genome, signal transduction, membrane transport, glycan biosynthesis and carbohydrate and polyketids and terpenoids metabolism were specifically enriched in the unique genome. An accessory genome was particularly rich in nucleotide and amino acids metabolism, replication, repair and drug resistance (Figure 11B).

### 3.9. SM Biosynthesis Abilities of the BA Pan, Core and Accessory Genomes

Bioinformatic tools for SM biosynthesis prediction such as antiSMASH 3.0 [30], NapDos [31], NP.search [32], and bacteriocin specific software BAGEL3 [33] were used to uncover secondary matabolite clusters from BA and other BC group 19 genomes. Figure 12A highlights the numerous and diverse SM clusters recovered with the different programs. Numerous unknown yet-to-be-discovered SM have been deciphered along with SM clusters encoding described products such as bacitracin, petrobactin and thurincin.

### 3.10. Predicted Natural Products Richness and Location within BA Genomes

A natural product survey in the core, accessory and unique BA genomes highlighted the location of secondary product clusters within the three genomes. Only petrobactin was present in the BA core genome while bacitracin and thurincin are located with the accessory and unique genomes, respectively (Figure 12B).

### 3.11. BA Genome Mining of Functional Genes for Detoxification of Organic Pollutants, Fitness and Competition

BA Whole genomes were mined for functional genes allowing detoxification of organic pollutants, fitness and/or competition. BA genomes proved to be endowed with a large battery of pathways for degradation of organic pollutants or xenobiotics such as styrene, xylene, dioxin, cyclohexane and benzoate (Figure 13). High transport and detoxification capacity of antibiotics was also observed.

They show large set of genes for antibiotic biosynthesis, biofilm formation, quorum sensing and chemotaxis (Figure 13A). BA accessory and unique genomes proved also rich in pathways for degradation of organic pollutants or xenobiotics (Figure 13B).

## 4. Discussion

Phosphogypsum, a by-product of the production of fertilizer from phosphate rock, is stored in large stacks. PG discharge into the sea has been practiced in Tunisia with dramatic consequences to marine ecosystems [2]. PG contains toxic components harmful to ecosystems and human health; these include heavy metals and radionuclides [1]. PG use as a fertilizer has also been limited by its toxic effect on plant growth and development and earthworms [3,4,5,6]. Its use as a construction material is also seriously limited by its radioactivity [39]. PG used in our study shows high metal content that has been linked to its ecotoxicity [4]. Our ICP-OES analysis shows different toxic metal concentrations according to PG provenance [7,8]. PG toxicity warrants the development of ecofriendly smart remediation means of the million tons piled on stacks. Here, we take advantage of PG and associated plant endophytic microbiomes to develop environmentally sound means for bioremediation of PG. In a first step, we analyzed the PG and SF, SM, MN and AI endophytic microbiomes by Illumina high throughput sequencing. Our results document more diverse bacterial microbiomes in PG than in plant endophyte microbiomes. The SF, SM, MN and AI endophytic bacterial microbiomes were significantly less diverse (low richness) and more equal (high evenness) than PG communities, a result that is in agreement with the study of Marasco et al. [40] on desert spear grasses. The PG bacterial community was dominated by firmicutes (39.3%), proteobacteria (24.2%) and actinobacteria (20.4%). This is in agreement with the study of Zouch et al. [8]. SF, SM, MN and AI endophytic microbiomes were dominated by protebacteria (89.6%) and firmicutes (10.4%); proteobacteria (98.5%); proteobacteria (96.7%) and proteobacteria (99.8%), respectively, in agreement with the reports of Gonzalez et al. [41] and Hassani et al. [42]. Similar results have also been reported for the microbiome of arugula (Eruca sativa Mill.) by Cernava et al. [43], *Vitis vinifera* [40], Apple [44] and *Trifolium pratense* [45]. For PG bacterial community bacilli (38.3%) dominated the firmicutes with main genera being *Bacillus* (20.7%) and *Enterococcus* (17.6%). Similar results have been reported by Kolekar et al. [46] for atrazine exposed soil in microcosm experiments. Alpha proteobacteria (18.8%) dominated the proteobacteria and actinobacteria were dominated by the genus *Kocuria* (15.2%), known to occur in diverse high pollution load wastes [47]. SF and SM endophytic microbiomes were dominated by Alpha proteobacteria (85.9% and 95.2%, respectively), while MN and AI endophytic microbiomes were dominated by Gamma proteobacteria (96.6% and 95.5%, respectively) with the main genera being Enterobacteriaceae genera (50.9%) undescribed genera and *Halomonas* (44.8%). All these genera have been frequently described as inhabitants of plant roots [44]. High specificity of taxa to their origin or host was observed (136), while only two taxa were common to PG and plant endophytic microbiomes. Network analysis revealed two main co-occurrences groups: a first group represented by *Clostridium gasigenes*, *Prevotella copri*, *Cesiribacter adanamensis* and *Shewanella algae* and a second group represented by *Paenibacillus aminolyticus*, *Carnobacterium viridans*, *Bacillus formainis*, *Pseudomonas fragi* and *Chromohalobacter salixigens*. These findings are of paramount importance to understand how bacterial consortia are recruited for efficient bio or phytoremediation. Metagenome functional prediction documented no clear difference between PG and plant endophytic microbiomes despite their diverse compositions. This result is in agreement with microbiomes being selected according to their functional content as suggested by Hassani et al. [42]. Along with the main functional categories, metabolism (49.4%), genetic information processing (16.3%), unclassified (14.8%) and environmental information processing (14.7%), xenobiotics, biodegradation and metabolism (2.3%), metabolism of terpenoids and polyketides (2%) and biosynthesis of other secondary metabolites (1%) are also represented. Similar findings were reported in Li et al. [48] for plant adaptation to adverse environments with high calcium contents. Given these findings, it was relevant to recover culturable population and to test single bacterial isolates metabolic capacities.

The culturable PG community was dominated by *Bacillus* spp. belonging mainly to the BC group. The BC group is widely linked to bioremediation [42,49] and the endophyte culturable microbiome of SF, SM.

*MN* and *AI* were more diverse than PG and lumped representants of 11 different genera. All of them are widely associated with a plant endophyte life style and efficient phytoremediation potential [10,15]. The PG and plant microbiomes were endowed with huge metabolic capacities. SF, SM, MN and AI endophytic communities have strong PGP activities reminiscing their endophytic life style [10]. While, PG and plant microbiomes displayed bioremediation and stress tolerance traits. Numerous studies reported similar results with different plants endophytes [15,43,50]. Interestingly, waste degradation potential was not limited to PG but extended to other recalcitrant wastes including TWW and OWW, insecticides such as imidacloprid, permethrin and dimethoate and the herbicide glyphosate. This result is of critical importance and is supposed to stimulate more studies into endogenous waste and associated plants microbiomes. Given the importance of these cultivable PG and associated plants endophyte microbiomes metabolic capacities, we decided to sequence the genomes of PG 1, PG 9, PG 17, PG 18 and PG 26 highly effective in the degradation of, TWW, PG, metals, insecticides and OWW.

The five sequenced species were all identified as BA using the gold standard for bacterial species identification based on phylogenomics, GGD and ANI inputs for definition of species boundaries [10,51]. Moreover, using the concatenated amino acid sequences of core genome derived from the five genomes of BA to reconstruct phylogenetic relationships among the BA genomes, we confirmed the results obtained using GGD and ANI analysis suggesting that isolates belong to BA.

BA has previously been isolated from the tailings of Panzhihua mining area and has high tolerance to vanadium (BA strain N35-10-2, GenBank accession: MAOE00000000) and Silt in tailing reservoir area (BA strain PFYN01, GenBank accession: CP034548). Comparative genomics analysis of BA PG 1, PG 9, PG 17, PG 18, PG 26, PFYN01 and N35-10-2 document their status as different strains of BA. These differences have been further confirmed by analysis of the different strains synteny and conserved and unique gene families. A species core genome, represented by gene sets shared by available genomes, describes the common metabolic and functional features of the species [10,15]. While pan genome mirrors full potentialities of considered species. Therefore, studying core and pan genome functional content is of paramount importance to decipher major intra species trends and main lifestyles [10].

Comparative genomics of BA isolates allowed showing that the species pan-genome might be an open pan-genome experiencing frequent changes through gene gains and losses or lateral gene transfers for efficient environmental adaptations [15]. Genome mining for detoxification and degradation of organic pollutants allowed conclusion that BA has an impressive molecular tool box for environmental bioremediation. Additionally, secondary metabolites have been suggested as relevant actors in the biodegradation of organic pollutants [10]. Genome mining of the PG BA isolates allowed to shed the light on relevant SM already reported for their bioremediation activity such as petrobactin [52]. Studying functional content of core and pan genome is of paramount importance to decipher major trends of intra species evolution and lifestyles [10,50,53]. In BA, while, the core genome was enriched with basic metabolic functions such as translation, ribosomal structure and biogenesis as well as coenzyme and nucleotide transport and metabolism, functional genes related to secondary metabolites biosynthesis, xenobiotic degradation, transport, and catabolism and defense mechanisms have been enriched in the accessory genome. A unique genome was enriched in cell wall, membrane and envelope biogenesis and replication recombination and repair along with genes of unknown or general function prediction only. We speculate that given the lifestyles of bacteria used in this study basic metabolic functions have been concentrated in the core genome while metabolic functions such as repair, membrane and envelope biogenesis, biosynthesis of SM and xenobiotic degradation have been allocated to accessory and unique genomes of strains to allow them to adapt to specific niches such PG in this study. More genome sampling is needed to infirm or confirm this hypothesis.

## 5. Conclusions

Despite the serious PG negative environmental impact involving drawbacks on human health, there is no bio/phytoremediation schemes to deal with this deleterious waste. PG and plant-associated endophytes microbiomes offer a credible alternative towards efficient management. A prerequisite for this is the characterization of these microbiomes and their functional investigation. Using metagenomic analysis and culture-based approaches, we describe unique communities thriving in this extreme environment. PG and plant-associated endophytes microbiomes proved to be endowed with an impressive battery of metabolic xenobiotic biodegradation capacities and genomic plasticity allowing them to adapt to these extreme environments. The results of this study will probably ultimately allow efficient development of bio/phytoremediation schemes of phosphogypsum.

## Figures and Tables

**Figure 1 microorganisms-07-00382-f001:**
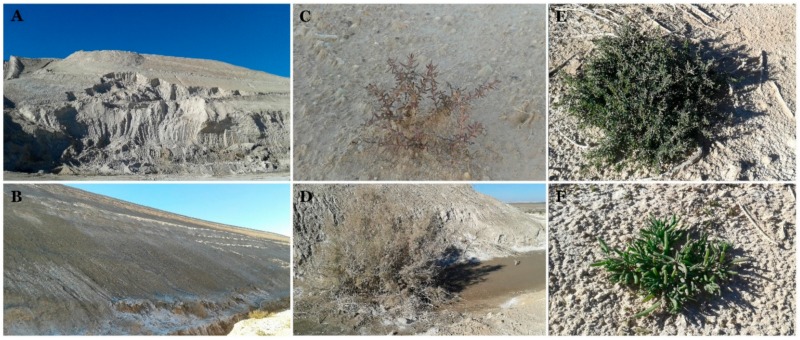
Phosphogypsum piles in CPG, Gafsa, Tunisia (**A**,**B**), *Arthrocnemum indicum*, *Suaeda fruticosa*, *Suaeda mollis* and *Mesembryanthmum nodiflorum* growing in areas contaminated with phosphogypsum in the vicinity of the PG piles (**C**–**F**).

**Figure 2 microorganisms-07-00382-f002:**
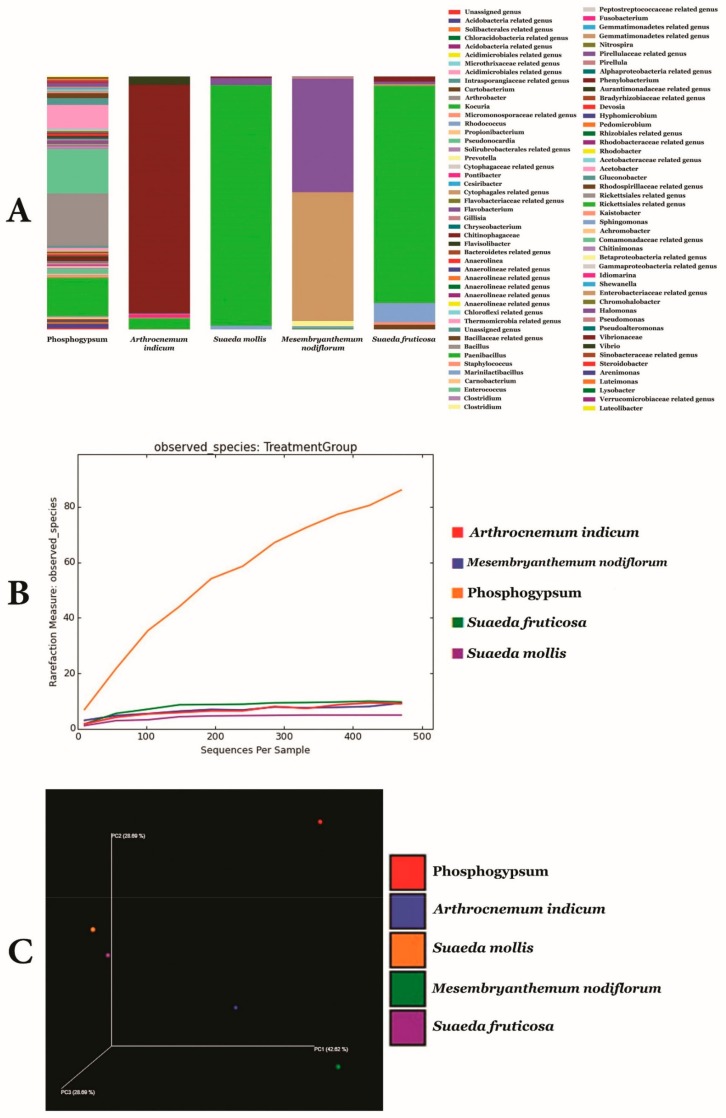
Genus level relative microbiome composition (**A**) Microbiome rarefaction curves (**B**) and principal coordinates analysis (**C**) of phosphogypsum and plant microbiomes generated using Mothur.

**Figure 3 microorganisms-07-00382-f003:**
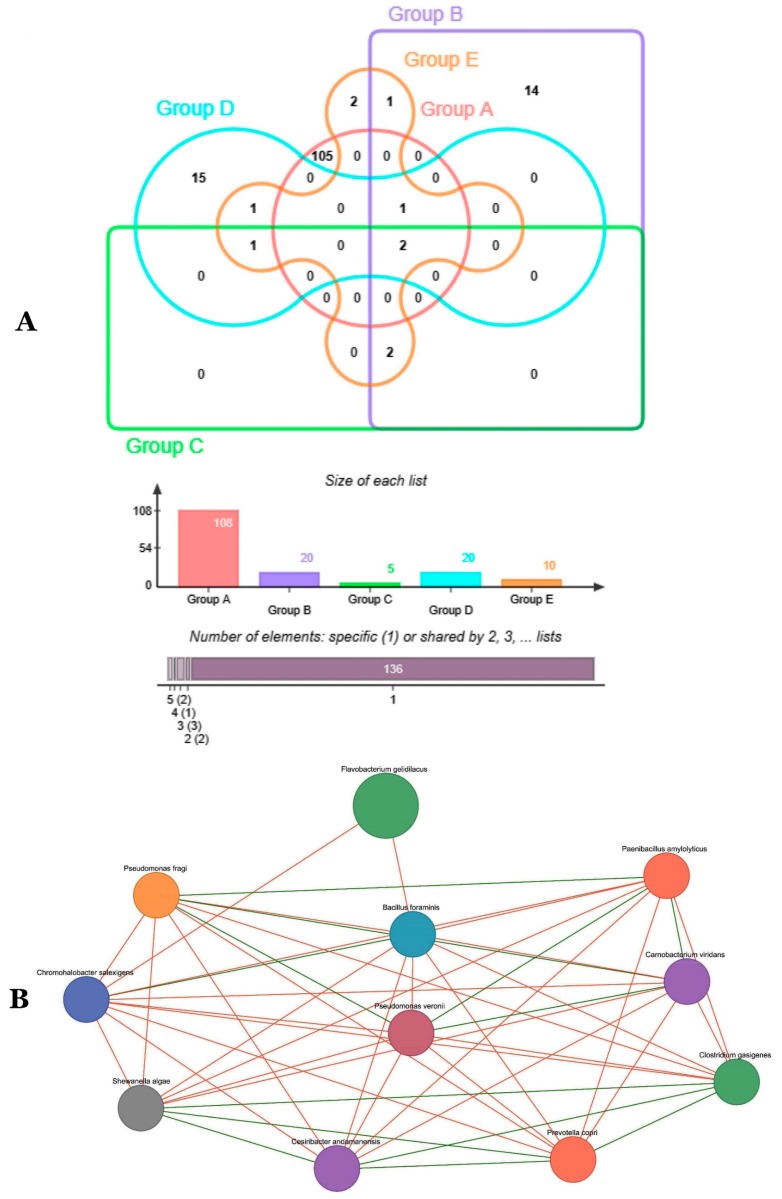
Venn diagram showing overlap between observed OTUs (**A**) and Co-occurrence network of key OTUs (**B**) for PG and plant microbiomes.

**Figure 4 microorganisms-07-00382-f004:**
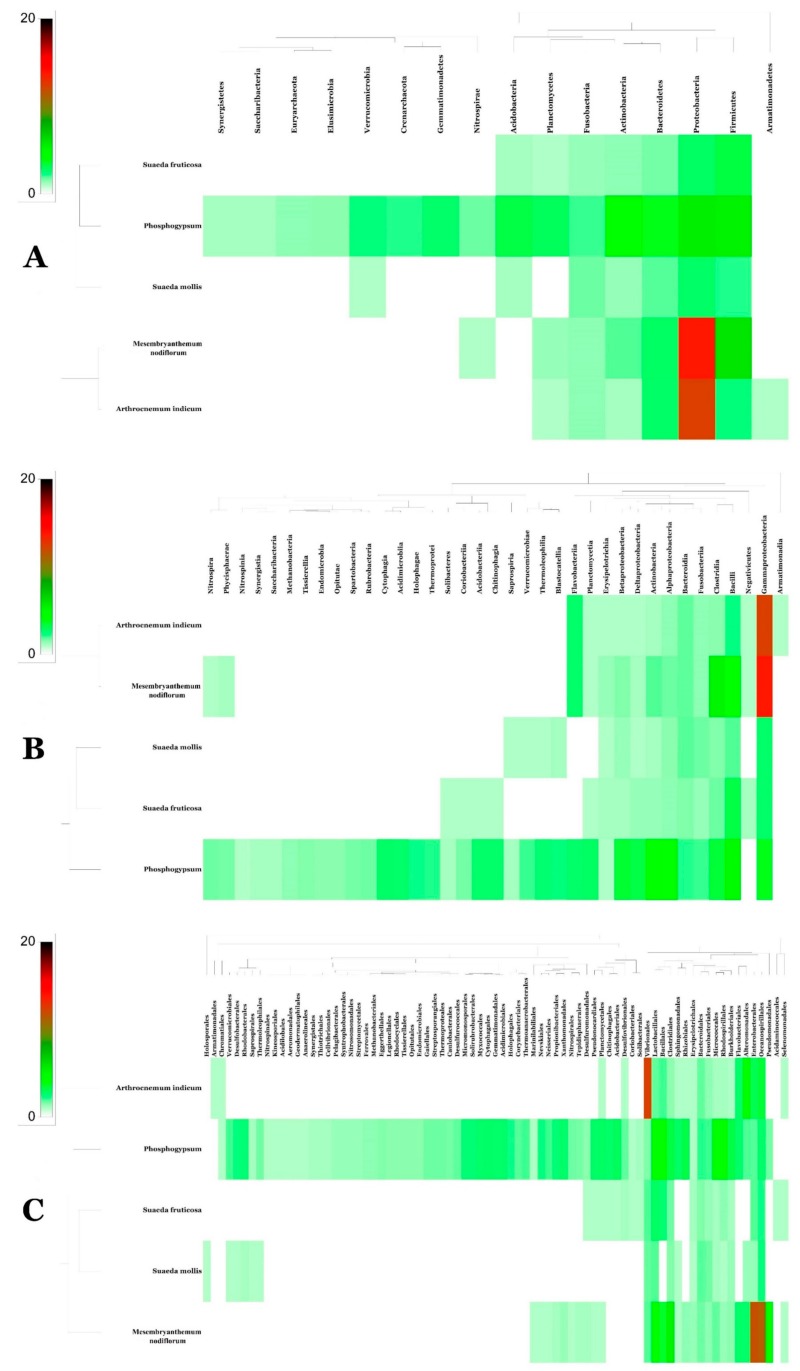
Bacterial community heat map of PG and plant microbiomes generated using SEED2 in phylum (**A**) order (**B**) and class (**C**) levels, respectively.

**Figure 5 microorganisms-07-00382-f005:**
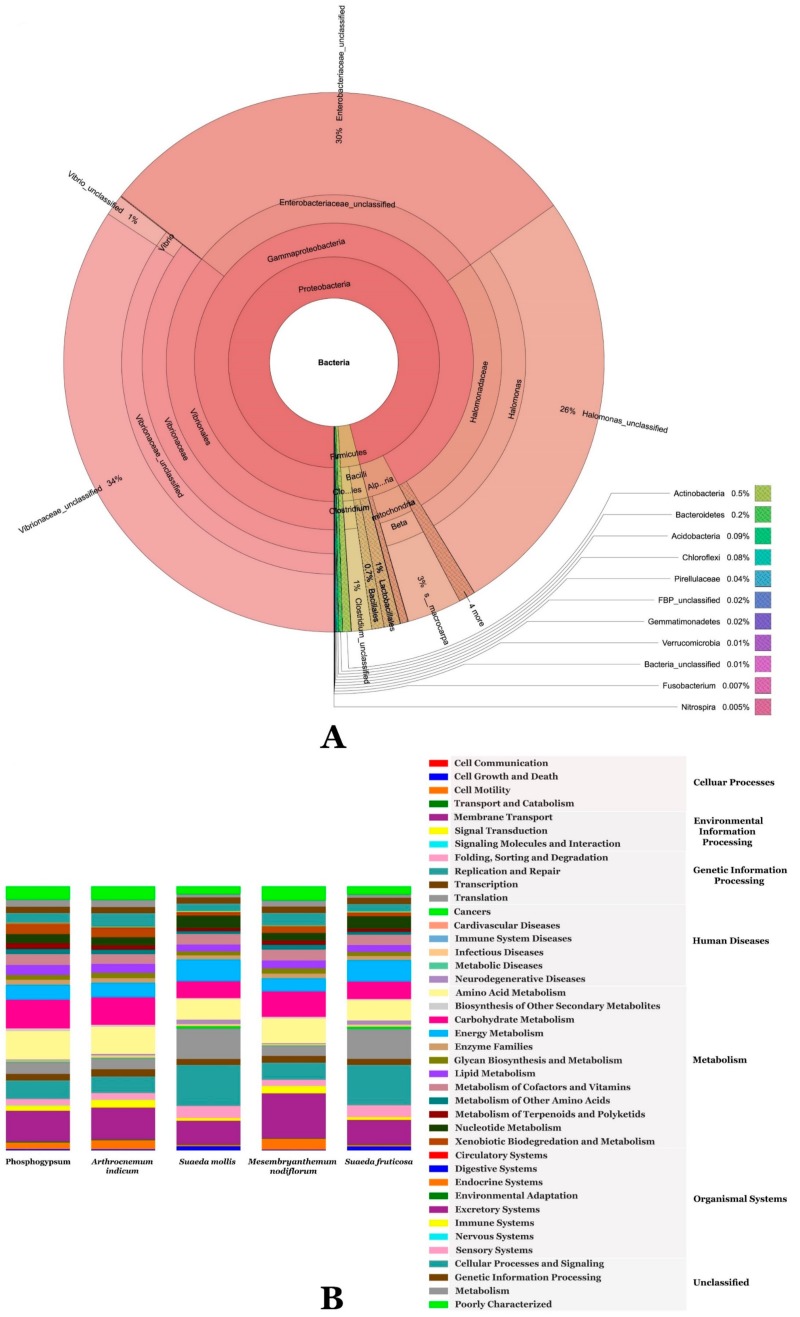
Krona chart (**A**) and PICRUSt classification of KEGG Orthologies (KO) (**B**) of phosphogypsum and plant microbiomes generated using Mothur.

**Figure 6 microorganisms-07-00382-f006:**
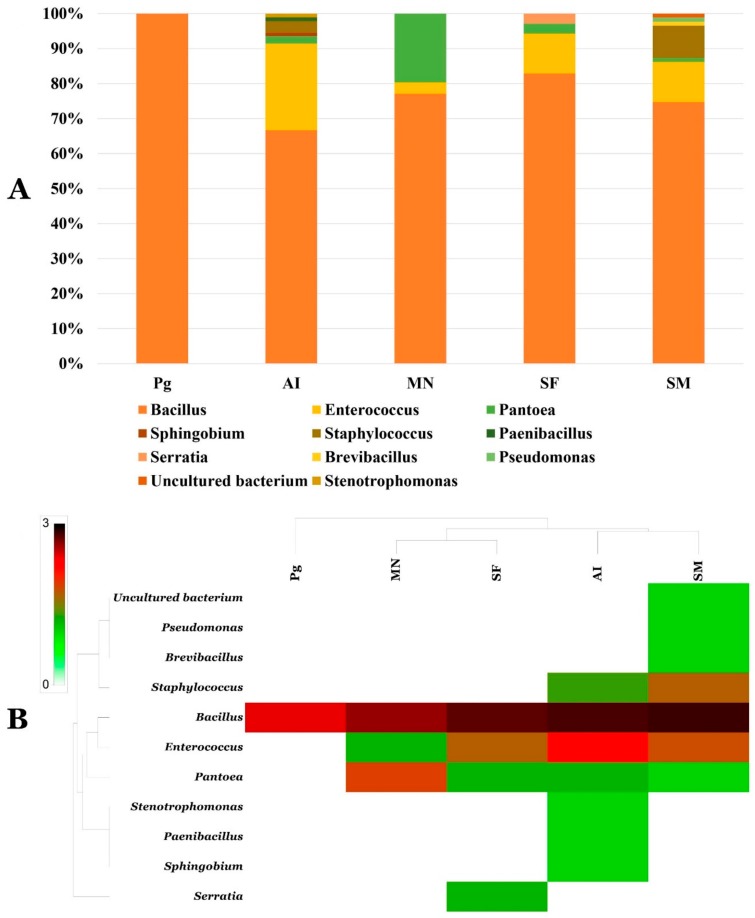
Bar chart showing diversity (**A**) and Heat-map of genera abundance of culturable Phosphogypsum and plant microbiomes (**B**).

**Figure 7 microorganisms-07-00382-f007:**
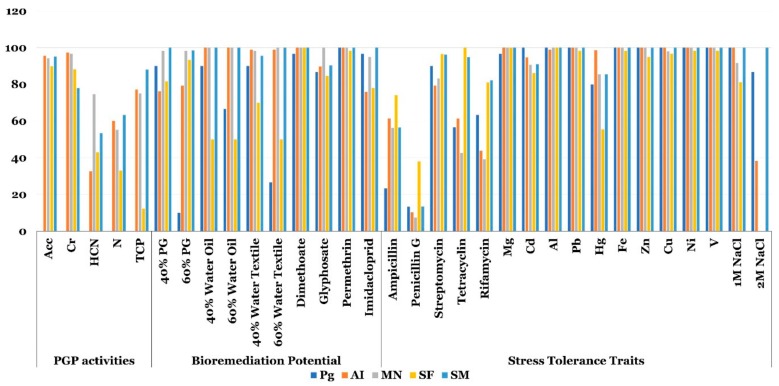
Bar chart showing PGP activities prevalence, bioremediation and stress tolerance traits of culturable Phosphogypsum and plant microbiomes.

**Figure 8 microorganisms-07-00382-f008:**
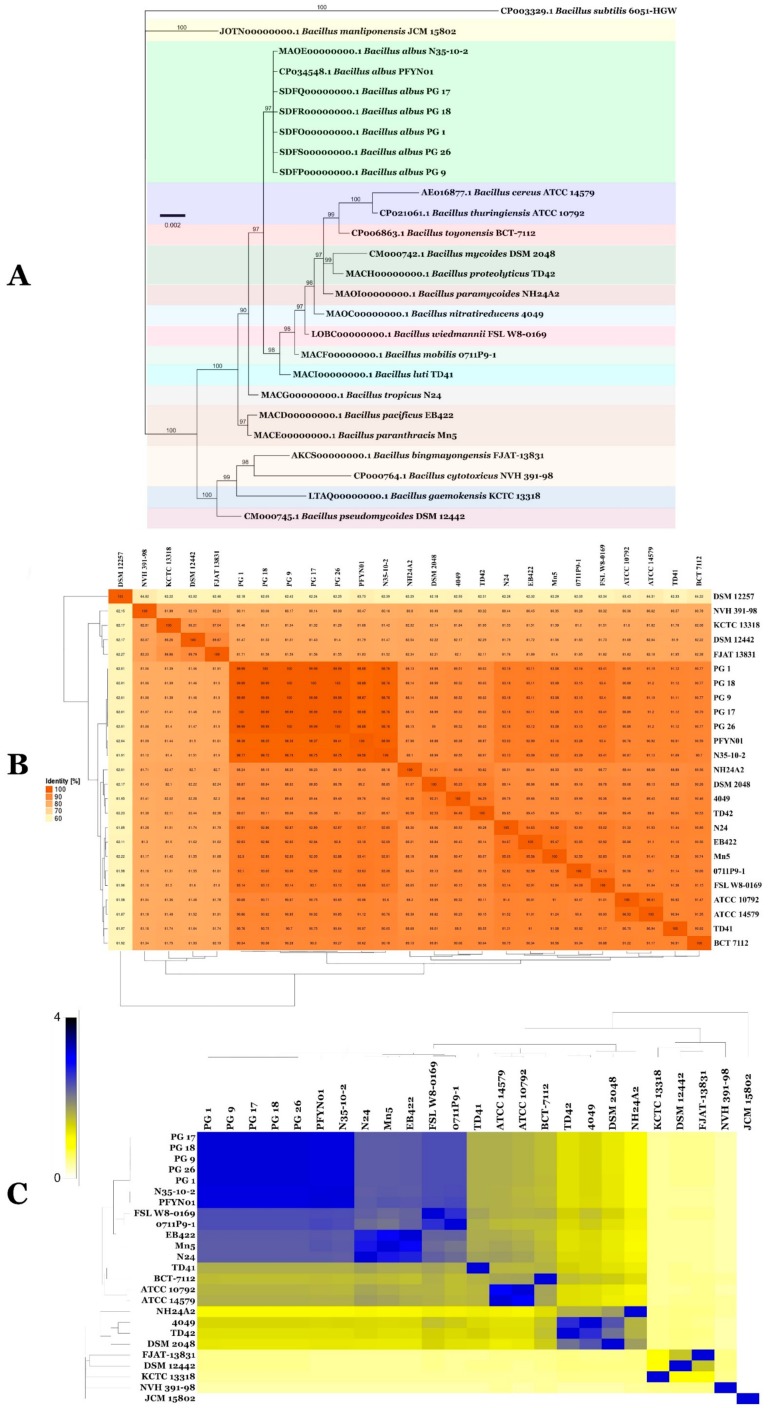
Bayesian inference phylogenomic tree of *Bacillus albus* and closest species (**A**). The tree was rooted with *B. subtilis* ATCC 6051-HGW. Bar, number of expected changes per site. ANI and GGDC values (**B**,**C**) between isolates generated using EzBiocloud and GGDC web services.

**Figure 9 microorganisms-07-00382-f009:**
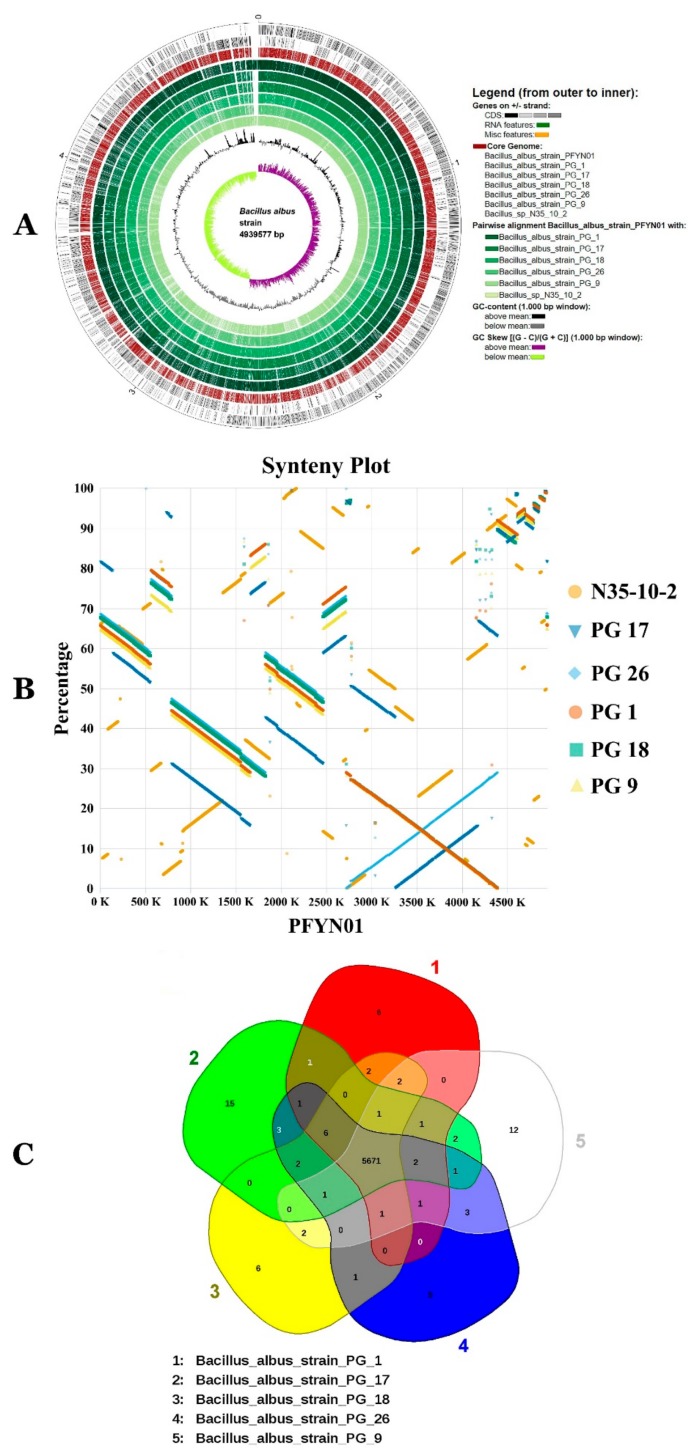
Circular genome comparison (**A**), synteny plot (**B**) and Venn diagram (**C**) showing overlap between common *Bacillus albus* strains genes.

**Figure 10 microorganisms-07-00382-f010:**
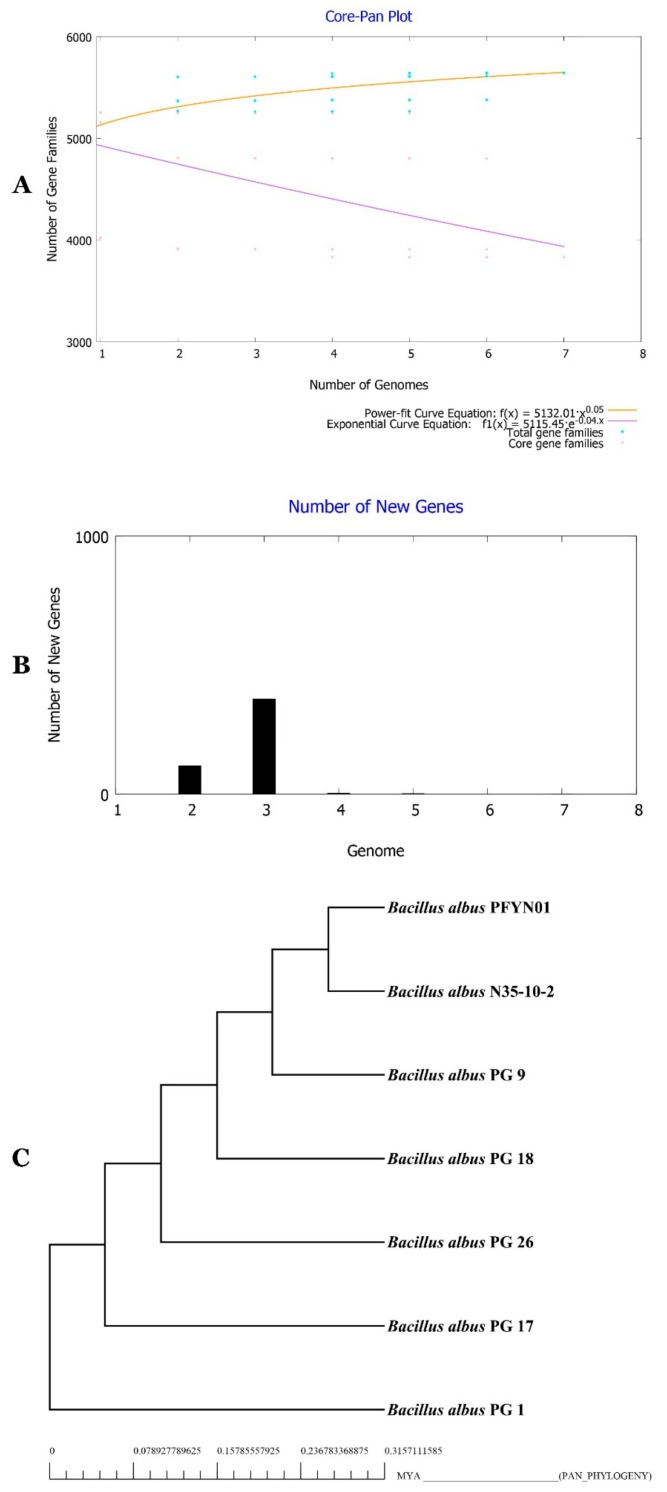
Core-Pan genome plot of BA isolates based on number of genomes and number of gene families (**A**), Number of new genes identified in genomes of BA (**B**), Tree of core-genome phylogeny of BA isolates (**C**).

**Figure 11 microorganisms-07-00382-f011:**
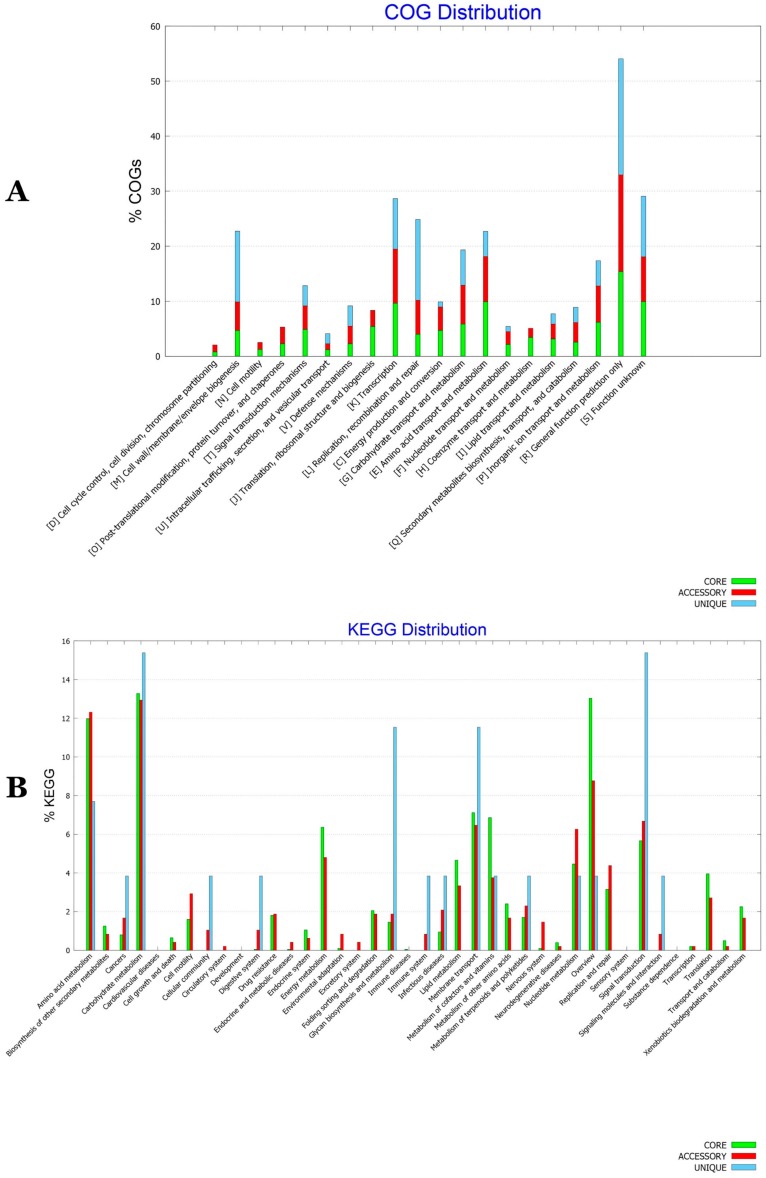
COG (**A**) and KEGG (**B**) distribution among core, accessory and unique genomes of *BA* isolates.

**Figure 12 microorganisms-07-00382-f012:**
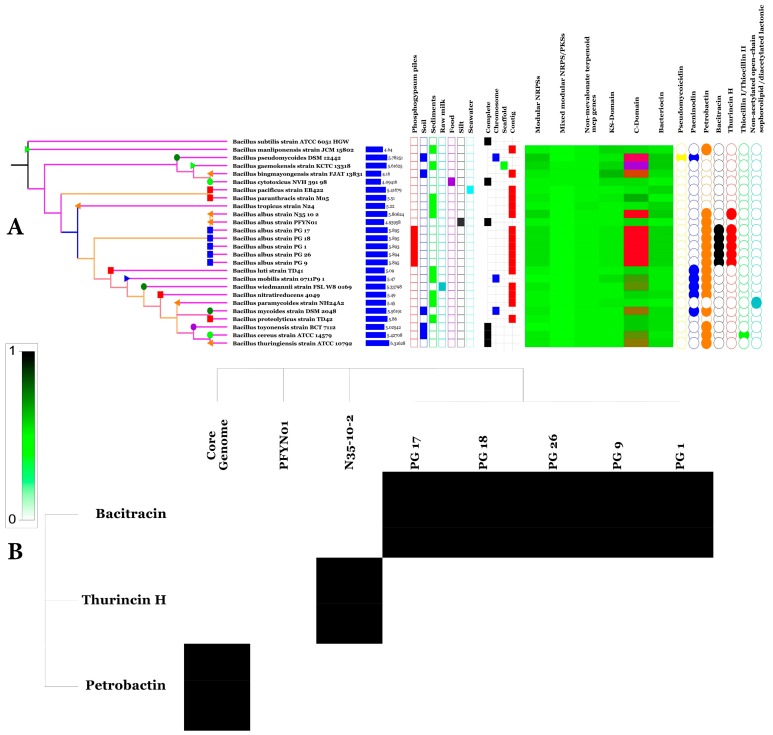
Chart presenting country of origin (
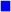
 Tunisia, 
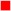
 Pacific of Ocean, 
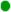
 the United States, 
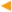
 China, 
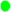
 France, 
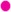
 Japan, 
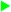
 South Korea) (**A**), genome size, isolation source, level of sequencing, prevalence of secondary metabolite clusters and types genes (from left to right), Heat map of secondary metabolite prevalence among *BA* core and accessory genomes (**B**).

**Figure 13 microorganisms-07-00382-f013:**
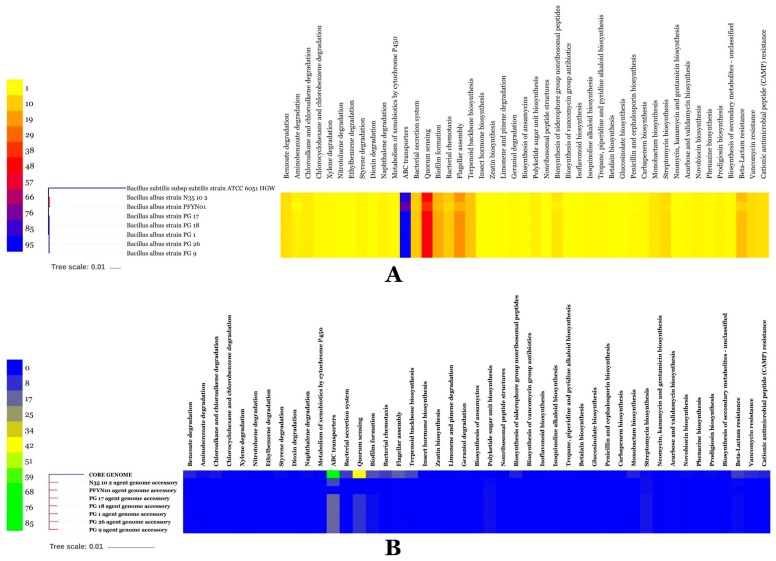
Heat map prevalence of genes contributing to bacterial fitness, organic pollutant degradation and stress tolerance in BA genomes (**A**), Heat map prevalence of genes contributing to bacterial fitness, organic pollutant degradation and stress tolerance in BA accessory genomes (**B**).

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
