# Peer review of "Metagenomic Insights and Genomic Analysis of Phosphogypsum and Its Associated Plant Endophytic Microbiomes Reveals Valuable Actors for Waste Bioremediation"

_microorganisms, 2019, doi:10.3390/microorganisms7100382_

Round 1

Reviewer 1 Report

Ben Mefteh et al., investigated the phosphogypsum microbiomes and plant-associated microbiomes by metagenomic analysis. In addition, PGP, salt resistance, PG bioremediation potentials were evaluated. Genomes of the five members of BA were further analysed. It is a nice idea to explore the PG and plant associated endophytes microbiomes and how this information can be used for PG bioremediation and efficient management. 
The manuscript merits publication, however, some rework are needed before acceptance.
In my opinion, the writing style should be improved, text should be improved for a smoother reading, and some results/statements are lacking background or precise description; the authors shall point out how these data could be (potentially) used for PG bioremediation, and some specific groups of microorganisms?
The following issues should be addressed:
Please consider to rename the title,
L102-108, please merge this two sections and describe clearly how the microbiomes were identified.
L119, there should be a brief description although details can be included in the Suppl.
L145, why suddenly a number of 56.9 mg? what is the purpose of this paragraph? the contents of the Phosphogypsum shall be shown in a Table.
L152-164, please reorganize the text: first data on phosphogypsum microbiome and then plants endophytic microbiomes.
The authors should note the correct use of Nomenclature for microorganisms (often neglected). e.g. L153-170, L200-209.
L156-158, The plants SF, SM, MN and AI endophytic microbiomes were all dominated by Proteobacteria.
L165, their=?
L181-183, please move the text to M&M
L212-214, please move the text to M&M
L215-219, L222-225, data (Figure or Table?) are missing
L247, why Section 3.6, what is the contribution, it is confusing that PG isolates were mentioned if no pre-description.
L289-292, please move the text to Introduction or Discussion
Fig. 8B and Fig. 12 need improvement for better reading

Author Response

Reviewer 1

Open Review

English language and style

( ) Extensive editing of English language and style required 
( ) Moderate English changes required 
(x) English language and style are fine/minor spell check required 
( ) I don't feel qualified to judge about the English language and style 

Yes

Can be improved

Must be improved

Not applicable

Does the introduction provide sufficient background and include all relevant references?

(x)

( )

( )

( )

Is the research design appropriate?

(x)

( )

( )

( )

Are the methods adequately described?

( )

( )

(x)

( )

Are the results clearly presented?

( )

( )

(x)

( )

Are the conclusions supported by the results?

( )

(x)

( )

( )

Comments and Suggestions for Authors

Ben Mefteh et al., investigated the phosphogypsum microbiomes and plant-associated microbiomes by metagenomic analysis. In addition, PGP, salt resistance, PG bioremediation potentials were evaluated. Genomes of the five members of BA were further analysed. It is a nice idea to explore the PG and plant associated endophytes microbiomes and how this information can be used for PG bioremediation and efficient management.

The manuscript merits publication, however, some rework are needed before acceptance.

Thanks to reviewer! we tried our best to fulfil reviewer recommendations.

In my opinion, the writing style should be improved, text should be improved for a smoother reading, and some results/statements are lacking background or precise description; the authors shall point out how these data could be (potentially) used for PG bioremediation, and some specific groups of microorganisms?

The following issues should be addressed:

Please consider to rename the title,

Thanks to reviewer, Reviewer recommendation has been fulfilled.

L102-108, please merge these two sections and describe clearly how the microbiomes were identified.

Thanks to reviewer, Reviewer recommendation has been fulfilled.

L119, there should be a brief description although details can be included in the Suppl.

Thanks to reviewer, Reviewer recommendation has been fulfilled.

L145, why suddenly a number of 56.9 mg? what is the purpose of this paragraph? the contents of the Phosphogypsum shall be shown in a Table.

56.9 is the dry weight of phosphogypsum this a commonly used parameter to characterize phosphogypsum. It is to allow comparison of phosphogypsum waste used across different studies. All phosphogypsum contents are shown in Supplementary Table S2.

L152-164, please reorganize the text: first data on phosphogypsum microbiome and then plants endophytic microbiomes.

Thanks to reviewer, Reviewer recommendations have been fulfilled.

The authors should note the correct use of Nomenclature for microorganisms (often neglected). e.g. L153-170, L200-209.

Thanks to reviewer, Reviewer recommendations have been fulfilled.

L156-158, The plants SF, SM, MN and AI endophytic microbiomes were all dominated by Proteobacteria.

Thanks to reviewer, Reviewer recommendation has been fulfilled.

L165, their=?

Thanks to reviewer, Reviewer recommendation has been fulfilled.

L181-183, please move the text to M&M

Thanks to reviewer, Reviewer recommendation has been fulfilled.

L212-214, please move the text to M&M

Thanks to reviewer, Reviewer recommendation has been fulfilled.

L215-219, L222-225, data (Figure or Table?) are missing

Thanks to reviewer, Reviewer recommendations have been fulfilled.

L247, why Section 3.6, what is the contribution, it is confusing that PG isolates were mentioned if no pre-description.

Section 3.6 is necessary and we prefer to keep it. PG isolates have all been described in supplementary Figure S3.

L289-292, please move the text to Introduction or Discussion                                    

Thanks to reviewer, Reviewer recommendation has been fulfilled.

Fig. 8B and Fig. 12 need improvement for better reading

Thanks to reviewer, Reviewer recommendations have been fulfilled.

Reviewer 2 Report

Review Mefteh et al.
============

This study covers a tremendous amount of data and analyses. Unfortunately, the paper is not well written, it lacks clear objectives, motivation and an overall structure. A lot of experimental details are only available from the supplementary, and a lot of these details are misleading/confusing. Besides, there are plenty of details still missing. The figures are all (more than 12 figures in the main text plus many more in the supplement) hard to read and the captions do not describe them properly.

I can not recommend this manuscript to be published in Microorganisms.

Abstract

L. 22: Illumina sequencing is misleading, do you refer to amplicon sequencing or genomics?

L. 31: Missing article: the ... arsenal.

L. 32: Missing article: tool box in the ...

L. 34: I think it would help to name exemplary metabolites.

Introduction

L. 58-62: These examples seem to be unrelated to PG and I do not see how they add value to the writing. Are there better examples with a more direct link to PG?

L. 74: "Illumins sequencing technology" - Please see my comment above.

For my taste the introduction does not sufficiently underscore the relevance of studying the PG microbiome. You talk about the harm of PG in the beginning, but that is all. Are there any previous studies about the microbial ecology of PG? How much of a problem is PG globally?

Materials and Methods

L. 81-90: What about these effluents? They are not mentioned in the introduction.

It is not acceptable that so many details are missing and referenced to the supplementary material. Key aspects of the experimental procedures should be included in the main text.

There are inconsistencies, you partially talk about rDNA, rRNA genes or "Ribosomal RNA from DNA", this is confusing. Stay consistent. Illumina library preparation is not properly described (e.g. what is an Illumina paired-end kit?). It is not clear whether sequencing adapters have been added via PCR or kit-based, It is not correct to call paired-end assembled sequences contigs. Contigs assembled in the course of (meta)genome assemblies. Paired-end assembly is not described? Which tool was used? Settings for used tools are not given (e.g. QC via prinseq). This is major deficit with respect to the carried out mothur, qiime and picrust analyses.

Results

3.1 Partially nouns are written with capital letters (e.g. Magnesium), partially with small letters (e.g. calcium, iron).

3.2 16S rRNA gene-based analysis are not equivalent to metagenomic analyses. Genus names should be in italic.

3.3 The results with respect to Xenobiotic breakdown are not described with enough detail.

3.5 What was the rationale behind applying these assays?

3.6 How much of the microbial community do your isolates reflect?

3.9-3.11 These data are only described superficially.

Discussion

The whole discussion is highly repetitive when compared to the results. What about potential bioremediation schemes? What did you learn? How do PG affected sites compare to e.g. to typical heavy metal contaminated sites? You do not reflect your results against available literature.

Conclusion

The conclusions are superficial, what are the main take homes from all the data you collected?

Figures

Figure 2 is not readable (the legend/color code), The color code is not easy on the eyes. 2C looks like default qiime output for 3D PCoAs.

Figure 3, panel A, the Venn diagram is rather messy and not easy to read. The bar chart of 3A shows redundant information. The caption does not explain the figure sufficiently, especially the network.

Figure 4, what does the color code represent, what about the dendrograms? Again the caption is not sufficient.

Figure 5, again no proper caption. What is point of showing "human diseases" among the functional categories?

You have more than 13 figures and I have a hard time to understand the relevance/meaning of most of them.

Author Response

Reviewer 2

Open Review

(x) I would not like to sign my review report

() I would like to sign my review report

English language and style

(x)Extensive editing of English language and style required

() Moderate English changes required

() English language and style are fine/minor spell check required

() I don't feel qualified to judge about the English language and style

Yes

Can be improved

Must be improved

Not applicable

Does the introduction provide sufficient background and include all relevant references?

( )

( )

(x)

( )

Is the research design appropriate?

( )

( )

(x)

( )

Are the methods adequately described?

( )

( )

(x)

( )

Are the results clearly presented?

( )

( )

(x)

( )

Are the conclusions supported by the results?

( )

( )

(x)

( )

Comments and Suggestions for Authors

Review Mefteh et al.

============

This study covers a tremendous amount of data and analyses. Unfortunately, the paper is not well written, it lacks clear objectives, motivation and an overall structure. A lot of experimental details are only available from the supplementary, and a lot of these details are misleading/confusing. Besides, there are plenty of details still missing. The figures are all (more than 12 figures in the main text plus many more in the supplement) hard to read and the captions do not describe them properly.

I cannot recommend this manuscript to be published in Microorganisms.

We tried all our best to address reviewer concerns and we hope we reached our objective.

Abstract

22: Illumina sequencing is misleading; do you refer to amplicon sequencing or genomics?

Thanks to reviewer, Reviewer recommendation has been fulfilled.

31: Missing article: the ... arsenal.

Thanks to reviewer, Reviewer recommendation has been fulfilled.

32: Missing article: tool box in the ...

Thanks to reviewer, Reviewer recommendation has been fulfilled.

34: I think it would help to name exemplary metabolites.

Thanks to reviewer, Reviewer recommendation has been fulfilled.

Introduction

58-62: These examples seem to be unrelated to PG and I do not see how they add value to the writing. Are there better examples with a more direct link to PG?

Studies in the field are very sparse and no studies directly linked to PG are available

74: "Illumins sequencing technology" - Please see my comment above.

Thanks to reviewer, Reviewer recommendation has been fulfilled.

For my taste the introduction does not sufficiently underscore the relevance of studying the PG microbiome. You talk about the harm of PG in the beginning, but that is all. Are there any previous studies about the microbial ecology of PG? How much of a problem is PG globally?

We tried to fulfil reviewer recommendations. Studies targeting microbial ecology are lacking which is one of the major strong points of this paper. PG is a serious problem in countries producing phosphate such as Algeria, Tunisia, Morocco, Spain and Portugal or countries having a strong phosphate industry such as Poland, the USA…

Materials and Methods

81-90: What about these effluents? They are not mentioned in the introduction.

The effluents have been just used to check if the bacterial community could be used to degrade another effluent than phosphogypsum. We believe there is no need to detail it in the introduction.

It is not acceptable that so many details are missing and referenced to the supplementary material. Key aspects of the experimental procedures should be included in the main text.

We believe the paper is more easily readable while keeping detailed experimental procedures in the supplementary material. We think that people who are looking for experimental procedures can just download the supplementary data.

There are inconsistencies, you partially talk about rDNA, rRNA genes or "Ribosomal RNA from DNA", this is confusing. Stay consistent. Illumina library preparation is not properly described (e.g. what is an Illumina paired-end kit?). It is not clear whether sequencing adapters have been added via PCR or kit-based, It is not correct to call paired-end assembled sequences contigs. Contigs assembled in the course of (meta)genome assemblies. Paired-end assembly is not described? Which tool was used? Settings for used tools are not given (e.g. QC via prinseq). This is major deficit with respect to the carried out mothur, qiime and picrust analyses.

Thanks to reviewer for raising this point, we made clear how we performed the experiments.

Results

3.1 Partially nouns are written with capital letters (e.g. Magnesium), partially with small letters (e.g. calcium, iron).

Thanks to reviewer, Reviewer recommendations have been fulfilled.

3.2 16S rRNA gene-based analysis are not equivalent to metagenomic analyses. Genus names should be in italic.

Thanks to reviewer, Reviewer recommendations have been fulfilled.

3.3 The results with respect to Xenobiotic breakdown are not described with enough detail.

Thanks to reviewer, Reviewer recommendations have been fulfilled.

3.5 What was the rationale behind applying these assays?

We think that since the PG bacterial population should be rich with members that can degrade other effluents and that they are tolerant to other stresses including metals and antibiotics. See Pal et al. (doi: 10.1016/bs.ampbs.2017.02.001).

3.6 How much of the microbial community do your isolates reflect?

Bacillus is the main genus present in our study and we believe that our isolates help to have an initial assessment in a lacking field of research.

3.9-3.11 These data are only described superficially.

Given the amount of data in the paper we favoured to not detail this part. Additionally, the discussion part cites one of our papers that detail it.

Discussion

The whole discussion is highly repetitive when compared to the results. What about potential bioremediation schemes? What did you learn? How do PG affected sites compare to e.g. to typical heavy metal contaminated sites? You do not reflect your results against available literature.

Potential bioremediation schemes are not the focus of our paper. Actually, a follow up study is ongoing where bioremediation schemes are tested. What we learn in the study is clearly stated in the discussion and the conclusion. We believe our work provided clues and bacterial communities that can be incorporated in any putative bioremediation scheme to deal with the waste. PG affected sites and their comparison to heavy metal contaminated sites is not the focus of our paper. We can not treat all these aspects in a single paper. Results have been widely reflected against available data. Unfortunately, there is not extensive data available in this field.

Conclusion

The conclusions are superficial, what are the main take homes from all the data you collected?

We clearly state in our conclusion that our work could be considered as a first step for phosphogypsum bioremediation. We really believe that our statements in the conclusion are easily reachable and a follow up study is actually conducted in our laboratory.

Figures

Figure 2 is not readable (the legend/color code), The color code is not easy on the eyes. 2C looks like default qiime output for 3D PCoAs.

Thanks to reviewer, Reviewer recommendations have been fulfilled.

Figure 3, panel A, the Venn diagram is rather messy and not easy to read. The bar chart of 3A shows redundant information. The caption does not explain the figure sufficiently, especially the network.

Thanks to reviewer, Reviewer recommendations have been fulfilled.

Figure 4, what does the color code represent, what about the dendrograms? Again the caption is not sufficient.

Thanks to reviewer, Reviewer recommendations have been fulfilled.

Figure 5, again no proper caption. What is point of showing "human diseases" among the functional categories?

This is an important point because numerous studies showed a clear link between presence of detoxification mechanisms and the prevalence of antibiotic resistance genes and human diseases.

You have more than 13 figures and I have a hard time to understand the relevance/meaning of most of them.

The amount of data is really important and of relevance for journal readers mainly those that have are from the field. We favour having a detailed analysis rather than synthetic figures that can blur the conclusions.